# A molecule inducing androgen receptor degradation and selectively targeting prostate cancer cells

Serge Auvin[1],*, Harun Öztürk[7],*, Yusuf T Abaci[7], Gisele Mautino[1], Florence Meyer-Losic[1], Florence Jollivet[2,3,4], Tarig Bashir[1], Hugues de Thé[2,3,4,5,6], Umut Sahin[2,3,4,7]

**Aberrant androgen signaling drives prostate cancer and is targeted by drugs that diminish androgen production or impede androgen–androgen receptor (AR) interaction. Clinical resistance arises from AR overexpression or ligand-independent constitutive activation, suggesting that complete AR elimination could be a novel therapeutic strategy in prostate cancers. IRC117539 is a new molecule that targets AR for proteasomal degradation. Exposure to IRC117539 promotes AR sumoylation and ubiquitination, reminiscent of therapy-induced PML/RARA degradation in acute promyelocytic leukemia. Critically, ex vivo, IRC117539-mediated AR degradation induces prostate cancer cell viability loss by inhibiting AR signaling, even in androgen-insensitive cells. This approach may be beneficial for castration-resistant prostate cancer, which remains a clinical issue. In xenograft models, IRC117539 is as potent as enzalutamide in impeding growth, albeit less efficient than expected from ex vivo studies. Unexpectedly, IRC117539 also behaves as a weak proteasome inhibitor, likely explaining its suboptimal efficacy in vivo. Our studies highlight the feasibility of AR targeting for degradation and off-target effects' importance in modulating drug activity in vivo.**

## Introduction

Prostate cancer is the most common type of cancer in men in the United States and accounts for 30,000 deaths annually (Center et al, 2012). The aberrant growth of malignant prostate tissue is dependent on androgen receptor (AR) signaling (Corbin & Ruiz-Echevarria, 2016; Pelekanou & Castanas, 2016). Whereas exogenous administration of androgens (i.e., testosterone) enhances prostate cancer proliferation, reducing testosterone levels halts cancer progression, a strategy known as androgen deprivation therapy (ADT) (Imamura &

Sadar, 2016; Narayanan et al, 2016). ADT can be achieved by castration either surgically, or by interfering pharmacologically with testosterone production. Nowadays, ADT is often complemented with the use of AR antagonists (i.e., bicalutamide and enzalutamide), which compete with testosterone for binding to AR (Chen et al, 2008; Leibowitz-Amit & Joshua, 2012; Helsen et al, 2014; Bambury & Scher, 2015). This combination strategy consequently achieves complete androgen blockade and is largely effective in short-term clinical management of prostate cancer. Yet, prostate cancer relapses almost invariably, giving rise to castration-resistant prostate cancer (CRPC) (Armstrong & Gao, 2015; Bambury & Rathkopf, 2016; Yap et al, 2016). CRPC is associated with reactivation of the AR signaling pathway, despite very low levels of circulating testosterone (Wyatt & Gleave, 2015). The most common molecular mechanism is AR overexpression, which results from either amplification of the AR gene locus at Xq12 (i.e., duplications and X-chromosome polysomy) or enhanced AR stabilization through reduced ubiquitination and degradation (Chen et al, 2004; Scher & Sawyers, 2005; Armstrong & Gao, 2015; Chandrasekar et al, 2015a, 2015b). Strikingly, prostate cancer cells with high levels of AR are rendered hypersensitive to even minimal amounts of circulating testosterone. Mutations in the AR gene can also confer resistance (Yuan et al, 2014; Karantanos et al, 2015). They generally occur in the ligand-binding domain (LBD) and render AR constitutively active. Some other mutations result in the activation of AR by different steroid hormones, including progesterone and cortisol or even by antagonist drugs (Grist et al, 2015). Finally, alternative splicing or aberrant proteolytic processing may generate AR variants that lack the C-terminal LBD and are constitutively active (Nakazawa et al, 2014; Lu et al, 2015; Caffo et al, 2016).

Like many other transcription factors, AR expression, function, and turnover are tightly regulated at multiple levels, including posttranslational modifications (Anbalagan et al, 2012; Coffey & Robson, 2012). AR can be poly-ubiquitinated by distinct ubiquitin E3 ligases, including CHIP, SPOP, MDM2, or SIAH2, all of which can promote its degradation by the proteasome (Qi et al, 2013, 2015; van

[1]Ipsen Innovation, Les Ulis, France    [2]Université de Paris, Hôpital St. Louis, Paris, France    [3]Institut National de la Santé et de la Recherche Médicale (INSERM) unité mixte de recherche (UMR) 944, Equipe labellisée par la Ligue Nationale contre le Cancer, Institut de Recherche St. Louis, Hôpital St. Louis, Paris, France    [4]Centre National de la Recherche Scientifique (CNRS) UMR 7212, Hôpital St. Louis, Paris, France    [5]Assistance publique – Hôpitaux de Paris, Service de Biochimie, Hôpital St. Louis, Paris, France    [6]College de France, PSL Research University, INSERM UMR 1050, CNRS UMR 7241, Paris, France    [7]Department of Molecular Biology and Genetics, Center for Life Sciences and Technologies, Bogazici University, Istanbul, Turkey

Correspondence: umut.sahin@boun.edu.tr
*Serge Auvin and Harun Öztürk contributed equally to this work.

der Steen et al, 2013; An et al, 2014; Sarkar et al, 2014). The N-terminal domain of AR can undergo small ubiquitin-like modifier (SUMO) conjugation, which eventually attenuates its transcriptional activity (Poukka et al, 2000; van der Steen et al, 2013; Sutinen et al, 2014; Wu et al, 2019; Zhang et al, 2019). In other settings, SUMO conjugation may initiate protein degradation (Lallemand-Breitenbach et al, 2001, 2008; Dassouki et al, 2015). In acute promyelocytic leukemia or adult T-cell lymphoma, arsenic-induced, SUMO-triggered ubiquitination and proteasomal destruction of driver oncoproteins (PML/RARA and Tax, respectively) were shown to be the underlying mechanism for therapy response (Lallemand-Breitenbach et al, 2008; Tatham et al, 2008; de The et al, 2012; Dassouki et al, 2015).

Current therapies aim at reducing testosterone levels or inhibiting testosterone-AR binding, whereas most therapy escape mechanisms in CRPC rely on altered AR expression or mutations.

Directly targeting AR for destruction may, thus, represent a promising approach in fighting therapy-resistant disease (Scher & Sawyers, 2005; Chen et al, 2008; Balbas et al, 2013; Watson et al, 2015). Here, we describe a molecule, which selectively induces AR destruction, resulting in the loss of prostate cancer cell viability ex vivo. This molecule, referred to as IRC117539, binds to AR, promotes a series of posttranslational modifications, including SUMO2/3 and ubiquitin conjugation, and mediates AR degradation by the proteasome. AR-negative prostate cancer cells are refractory to IRC117539, arguing that drug-induced AR loss initiates prostate cancer cell death. Remarkably, IRC117539 is also effective in inducing AR destruction and cell death in androgen-insensitive, AR-positive prostate cancer cells, which may constitute an alternative treatment strategy for CRPC. Subsequently, we found that IRC117539 also behaves as a weak proteasome inhibitor, which may explain its

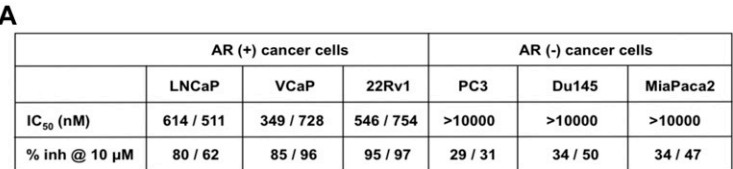

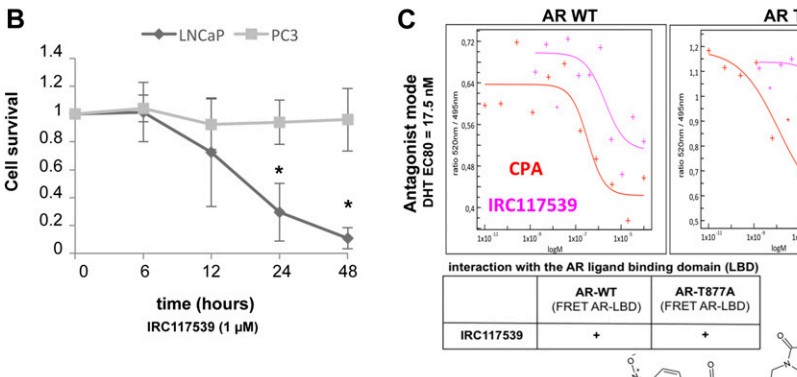

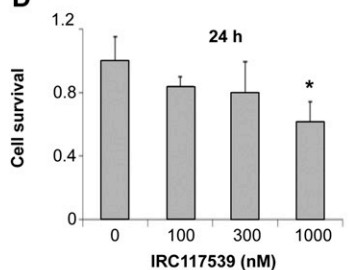

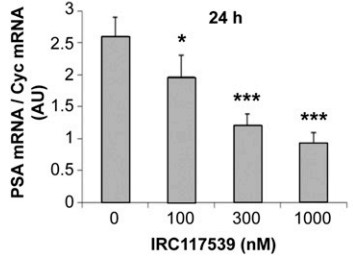

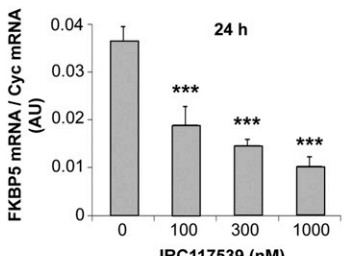

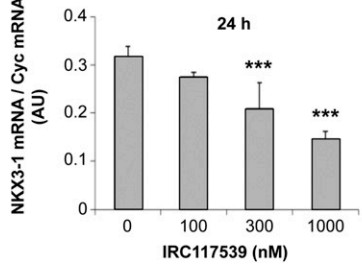

**Figure 1. The compound IRC117539 inhibits AR signaling axis and selectively induces loss of viability in AR-positive prostate cancer cells.**
**(A)** The effect and specificity of IRC117539 on prostate cancer cell proliferation. Three AR-positive prostate cancer and three different AR-negative cancer cell lines are shown for comparison. **(B)** Kinetics of IRC117539-induced loss of viability in AR-dependent LNCaP prostate cancer cells. AR-negative PC3 cells are also shown for comparison (IRC117539: 1 μM). **(C)** Testing antagonist mode of IRC117539 (chemical structure is shown) by FRET assay. The assay was performed as described in the Materials and Methods section using cyproterone acetate (CPA) as a positive control. IRC117539 binds AR in the LBD, both wt and T877A. **(D)** Dose-dependent reversal of AR target gene expression in LNCaP prostate cancer cells by IRC117539. FKBP5, PSA, and NKX3-1 mRNA levels were analyzed by Q-PCR and normalized to Cyc mRNA expression. Upper left graph shows cell viability at indicated doses of IRC117539. Data information: in (B and D), data are presented as mean ± SEM (n = 3). Asterisks denote statistical significance (*$P < 0.05$, ***$P < 0.001$, using $t$ test assuming unequal variances).

suboptimal efficacy in vivo in clearing AR. In line with this, pharmacologically boosting proteasome activity increases IRC117539's potency in inducing AR degradation, at least ex vivo. Our studies demonstrate the feasibility of promoting degradation of AR and highlight the importance of understanding off-target effects, as these may antagonize the desired activity.

# Results

### IRC117539 induces selective loss of viability in AR-positive prostate cancer cell lines

IRC117539 is a rationally designed and optimized compound based on a dimeric arylhydantoin motif that is known to bind to the AR LBD

(Figs 1C and S1A). A 6-d-long treatment with IRC117539 dramatically reduced the survival of cultured LNCaP, VCaP, and 22Rv1 prostate cancer cell lines whose proliferation was previously reported to be AR-driven (Fig 1A and B). Of particular interest, 22Rv1 cells are known to be androgen-insensitive because of expression of an AR isoform lacking the C-terminal LBD (Dehm et al, 2008; Cunningham & You, 2015) (ΔLBD in Fig 2A). The $IC_{50}$ of IRC117539 was determined to be in the low micromolar range. On the other hand, the survival of AR-negative PC3 and Du145 prostate cancer, MiaPaca2 pancreatic carcinoma, HeLa cervical cancer, or WI38 human fibroblast cells was only slightly affected (Figs 1A and B, and S1B). IRC117539 binds the AR LBD and displays AR antagonist activity (Fig 1C). In agreement with these findings, the molecule reversed the expression profile of several AR target genes (Fig 1D) and potently antagonized androgen (dihydrotestosterone)-induced LNCaP cell proliferation in a

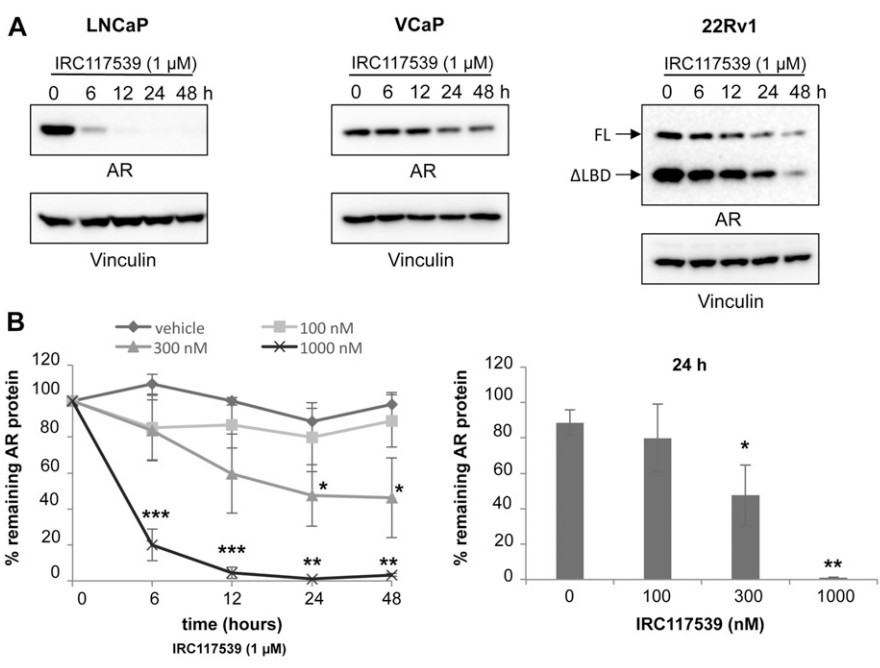

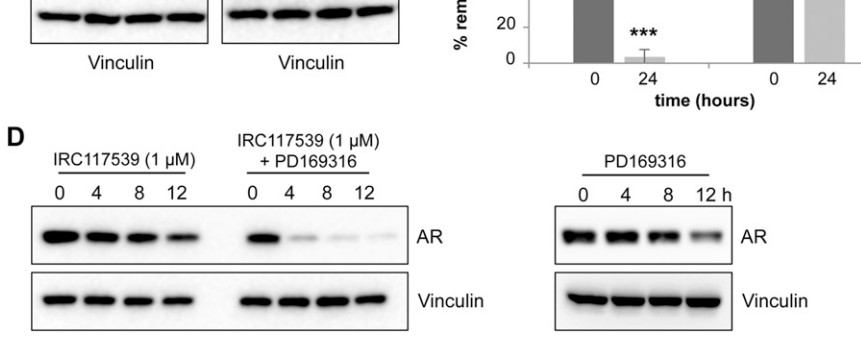

**Figure 2. IRC117539 induces AR protein degradation by the proteasome.**
**(A)** Western blot analysis of AR protein levels in LNCaP, VCaP, and 22Rv1 prostate cancer cells treated with 1 μM IRC117539 compound for indicated times. Note two different AR isoforms (FL, full-length retaining LBD; ΔLBD, the truncated variant) in 22Rv1 cells, which are both degraded. **(B)** Dose-dependent kinetics of IRC117539-induced AR degradation in LNCaP cells (left panel). AR protein levels were quantified after Western blot and normalized to vinculin (n = 3, representative blots are shown in Figs 2A and S2B). A dose–response graphic for the 24-h time point is shown in the right panel. **(C)** Proteasome-dependent degradation of AR. Western blot (left) shows AR protein levels in LNCaP cells exposed to 1 μM IRC117539, with or without proteasome inhibition. Right panel shows quantification of AR protein levels at 24 h (n = 3). **(D)** Western blot analysis of AR protein levels in LNCaP cells co-treated with 10 μM PD169316, a small molecule proteasome activator. Note accelerated kinetics of IRC117539-induced AR degradation. Data information: in (B and C), data are presented as mean ± SEM (n = 3). Asterisks denote statistical significance (*P < 0.05, **P < 0.01, ***P < 0.001, using t test assuming unequal variances).

dose-dependent manner (data not shown), demonstrating its inhibitory effect on androgen signaling. In summary, IRC117539 acts as an AR antagonist, which impairs AR-driven proliferation and survival of AR-positive prostate cancer cells in vitro.

## IRC117539 induces sumoylation and proteasome-dependent degradation of the AR

Inhibition of the AR signaling axis and AR target gene expression, as well as the subsequent loss of viability in AR-positive prostate cancer cell lines suggested that treatment with IRC117539 targets AR. Remarkably, exposure to 1 μM IRC117539 resulted in a sharp decline in AR protein levels in LNCaP, VCaP, and 22Rv1 cells, without affecting its mRNA production (Figs 2A and B, and S2A and B). IRC117539 actually diminished the levels of both full-length and ΔLBD AR variant in 22Rv1 cells. At this dose, AR protein loss was almost complete after compound treatment for 24 h and was reversed by simultaneous treatment with the proteasome inhibitor MG132, suggesting that IRC117539 induced proteasomal degradation of AR (Fig 2C). PD169316 is a small molecule proteasome activator recently reported to enhance the activity of both β1/5 and β2 subunits of the proteasome (Leestemaker et al, 2017). We observed that co-treatment with PD169316 increased the rate of IRC117539-induced AR destruction, dramatically boosting IRC117539's potency (Fig 2D).

Proximity ligation assays (PLAs) indicated that IRC117539 promoted physical association between AR and the proteasome (Fig 3A). To better understand the mechanisms by which AR is targeted to the proteasome, we used PLA to probe AR's posttranslational modifications (Fig 3A). In untreated cells, we found AR to be modified by SUMO1, but not SUMO2/3. Interestingly, upon treatment with IRC117539, there was a significant shift from SUMO1 to SUMO2/3 PLA signal amplification. In addition, IRC117539 exposure caused an increase in AR ubiquitin modification (Fig S3A). Immunoprecipitation analyses (performed on endogenous AR in LNCaP cells or AR transiently expressed in HeLa cells) confirmed that IRC117539 treatment induced massive conjugation of this protein by SUMO2/3 and ubiquitin (Figs 3B and S3B). Both SUMO1-conjugated AR (in untreated and treated cells) and SUMO2/3-conjugated AR (in treated cells) were largely nuclear, whereas ubiquitinated AR was found both in the nucleus and cytoplasm, as were AR proteasome PLA signals (Figs 3A and S3A). Interestingly, IRC117539 promoted physical interaction between AR and promyelocytic leukemia (PML) protein (Fig 3A). PML was shown to function specifically in promoting sumoylation and degradation of a subset of proteins (Guo et al, 2014; Sahin et al, 2014a, 2014b, 2014c, 2014d). Indeed, we found that AR possessed seven SUMO-interacting motifs (SIMs), previously shown to facilitate interaction with PML (Fig S3C) (Sahin et al, 2014c). To assess the role of global sumoylation in drug-induced AR degradation, we used ML792, a small molecule selective SUMO-activating enzyme inhibitor that blocks protein sumoylation (He et al, 2017). Co-treatment with ML792 dramatically impaired IRC117539-induced AR degradation, suggesting that the latter may be driven by sumoylation and ubiquitination of AR (Fig 3C).

Collectively, our results suggest that IRC117539 antagonizes AR signaling by specifically targeting and binding AR, and tagging it for degradation by the proteasome, most likely through a sequence of

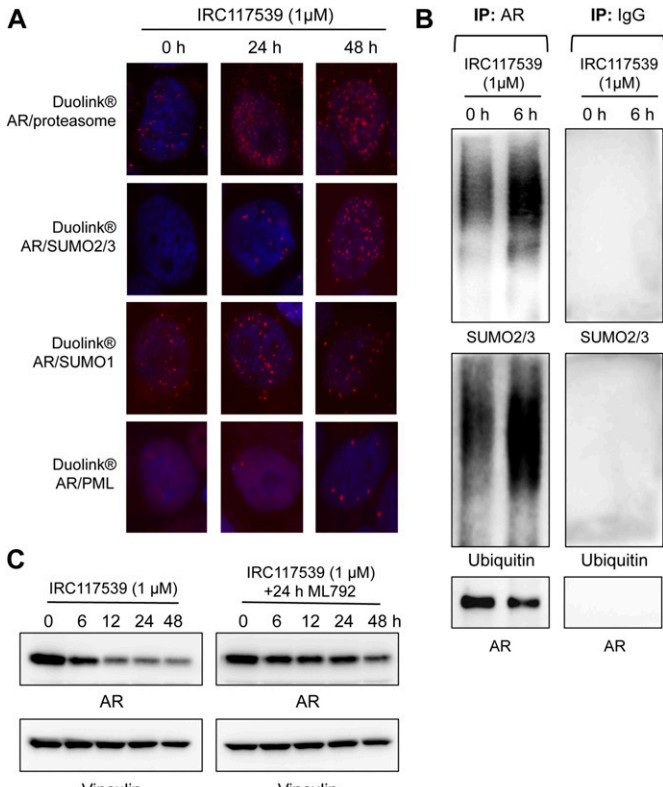

**Figure 3. IRC117539 induces AR posttranslational modifications before degradation.**
**(A)** Proximity ligation (Duolink) analyses probe AR physical interaction with the proteasome (20S particle) or with PML, as well as AR modification by SUMO1 or SUMO2/3 peptides at indicated times of IRC117539 treatment (1 μM) in LNCaP cells. **(B)** Immunoprecipitation of endogenous AR from LNCaP cells confirms IRC117539-induced AR conjugation by SUMO2/3 and ubiquitin. IRC117539 was used at 1 μM. A nonspecific IgG was used as a negative control (right panel). **(C)** Sumoylation is required for IRC117539-induced AR degradation. AR protein levels were assessed by Western blot in LNCaP cells co-treated with ML792 (1 μM), a small molecule sumoylation inhibitor.

posttranslational modifications involving SUMO2/3 and ubiquitin. This catabolic pathway is strikingly similar to the recently uncovered mechanism of therapy-induced PML/RARA degradation in acute promyelocytic leukemia and therapy-induced Tax oncoprotein degradation in adult T-cell lymphoma (Lallemand-Breitenbach et al, 2008; Tatham et al, 2008; Sahin et al, 2014c; Dassouki et al, 2015).

## IRC117539 also functions as a weak proteasome inhibitor

In addition to triggering AR conjugation by SUMO2/3, ubiquitination, and degradation, IRC117539 also promoted global accumulation of ubiquitin and SUMO conjugates in AR-positive (LNCaP and VCaP) and AR-negative (PC3 and Du145) prostate cancer lines, as well as in most other cell types (i.e., HeLa and WI38) and in vivo in mice (Figs 4A and B, S4A, B, D, and S5 and data not shown). SUMO2/3-modified proteins were affected to a greater extent in primary cells in comparison with SUMO1 conjugates (Fig S5 and data not shown). These observations raise the possibility that IRC117539 may be a general modulator of the ubiquitin/proteasome pathway, by

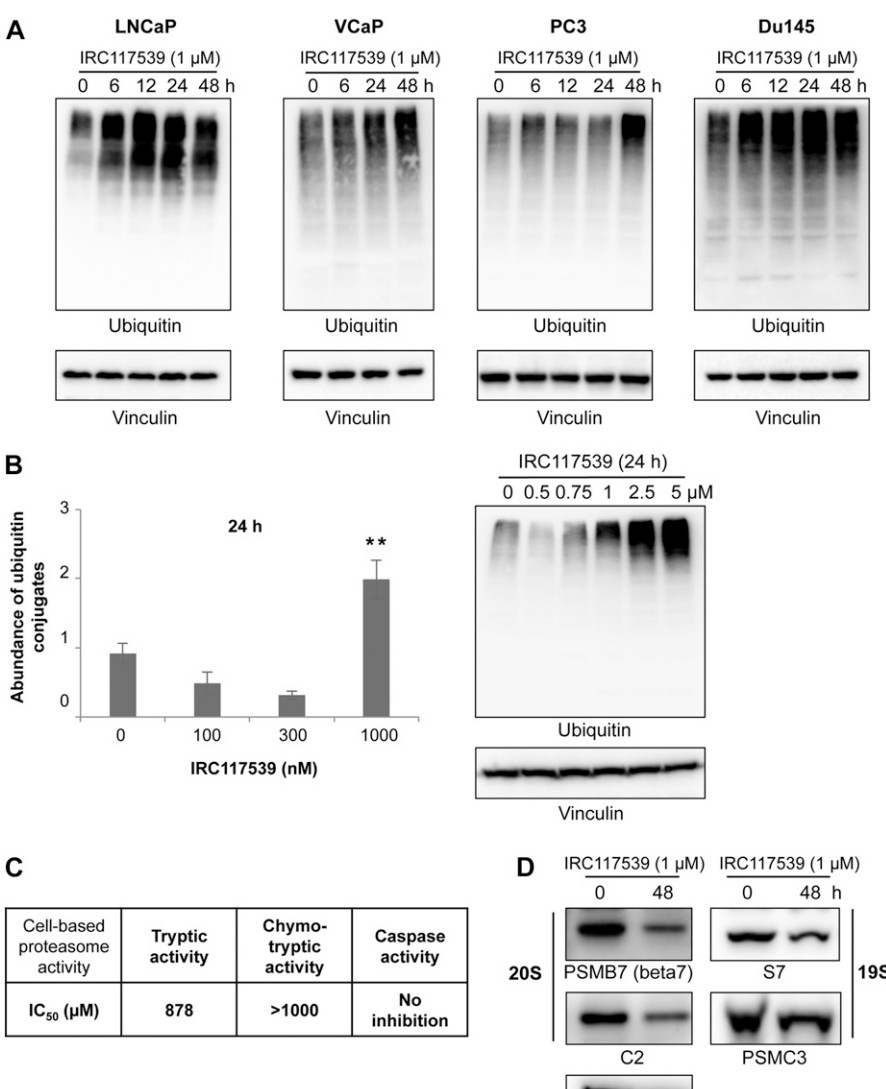

**Figure 4. IRC117539 impairs proteasome function.**
**(A)** Accumulation of global ubiquitin conjugates in AR-positive (LNCaP and VCaP) and AR-negative (PC3 and Du145) cells treated with 1 μM IRC117539. **(B)** Dose-dependent accumulation of ubiquitinated proteins in IRC117539-treated LNCaP cells at 24 h. Global protein ubiquitination was quantified after Western blot and normalized to vinculin (n = 3, representative blots are shown in right panel, in Fig 4A and in Fig S4A). The slight decrease in cellular ubiquitination at 100 nM, 300 nM, and 500 nM is likely due to enhanced proteasome activity and consistent with biphasic/hormetic cellular response (hormesis) to lower doses of a drug (Calabrese, 2008; Kendig et al, 2010). Data are presented as mean ± SEM. Asterisks denote statistical significance (**P < 0.01, compared with the 0 nM bar, using t test assuming unequal variances). **(C)** IRC117539's cell-based IC$_{50}$ values are indicated for proteasome's tryptic and chymotryptic activities. Analysis was performed in LNCaP cells. **(D)** Western blot analysis of various proteasome subunits in LNCaP cells treated with 1 μM IRC117539.

enhancing conjugation of ubiquitin-like modifiers or by inhibiting the turnover of SUMO(2/3)ylated and ubiquitinated proteins at the proteasome. To distinguish between these two possibilities, we examined the stability of short- or long-lived proteins after exposure to the compound. We observed a strong time-dependent accumulation of p53 and Mdm2, two short-lived proteins (Fig S5). On the other hand, the amount of PML, a stable protein with a half-life of 18 h, remained largely unchanged. Such an accumulation of short-lived proteins was indicative of a dysfunction of the proteasome rather than enhanced ubiquitin/SUMO conjugation. Indeed, stabilization of p53 was also observed in AR-negative cells (i.e., HeLa), suggesting that p53 accumulation was not a direct consequence of oncoprotein loss in tumor cells. Interestingly, we could not detect signs of proteasome inhibition in 22Rv1 cells (Fig S4C, see the Discussion section). Using cell-based proteasome activity assays, we found that IRC117539 inhibited both tryptic and chymotryptic activities of the proteasome (Fig 4C), although not as

efficiently as the proteasome inhibitor MG132 (ubiquitin blots in Fig S6C, and data not shown). In cell-based assays, the IC$_{50}$ of IRC117539 on the proteasome was determined to be in the low millimolar range (IC$_{50(tryptic\ activity)}$ = 878 μM, IC$_{50(chymotryptic\ activity)}$ > 1 mM). The caspase-like activity of the proteasome was not affected by IRC117539. Critically, protein expression of the catalytic β7, which contains trypsin-like activity, and structural C2 subunits of the 20S particle, sharply declined in LNCaP cells treated with IRC117539 (Fig 4D). Quantitative RT-PCR analyses demonstrated that transcription of genes encoding proteasome subunits β7 and C2 remained largely unchanged upon treatment (Fig S6A). Conversely, in cycloheximide chase experiments, both proteasome subunits β7 and C2 displayed reduced half-lives in the presence of IRC117539 (Fig S6B). Thus, these data suggest that IRC117539, aside from promoting AR degradation, also weakly impairs the function of the proteasome.

We finally analyzed the expression levels of CHOP and BiP, two commonly used markers for ER stress, which mediate cellular

unfolded protein response. Analysis of CHOP and BiP expression did not indicate a substantial rise in ER stress in various cells exposed to IRC117539 (Fig S6C and data not shown). Although CHOP levels were slightly up-regulated in LNCaP cells after a 24-h-long treatment with IRC117539, this effect was substantially weaker compared with MG132, a potent ER stress inducer, and tended to alleviate at longer hours of treatment (Fig S6C and data not shown).

### In vivo potency of the compound IRC117539

We next evaluated the in vivo potency of IRC117539 by performing Hershberger analyses in rats. This model assesses changes in the weight of androgen-dependent tissues in castrated male rats upon treatment with androgen agonists or antagonists (Kang et al, 2004). As shown in Fig 5A, we observed a significant decrease (75%) both in ventral prostate weight and in probasin expression in a Hershberger rat (castrated and supplemented in testosterone at 0.4 mg/kg) treated for 10 d with IRC117539. In addition, in mouse xenograft models transplanted with LNCaP prostate cancer cells, IRC117539 elicited a significant reduction (42%) in tumor volume, similar to enzalutamide, although the latter showed an earlier onset of efficacy (Fig 5B). In castration-resistant orthotopic LNCaP xenograft models, IRC117539 treatment did not significantly decrease AR protein levels (as determined by immunohistochemistry and ELISA, Fig 5C) and neither diminished tumor weight nor was it capable of reducing prostate specific antigen (PSA) levels in peripheral blood (Fig S7). Collectively, despite its promising in vitro activity, IRC117539's in vivo potency in decreasing AR protein levels, and in achieving sufficient

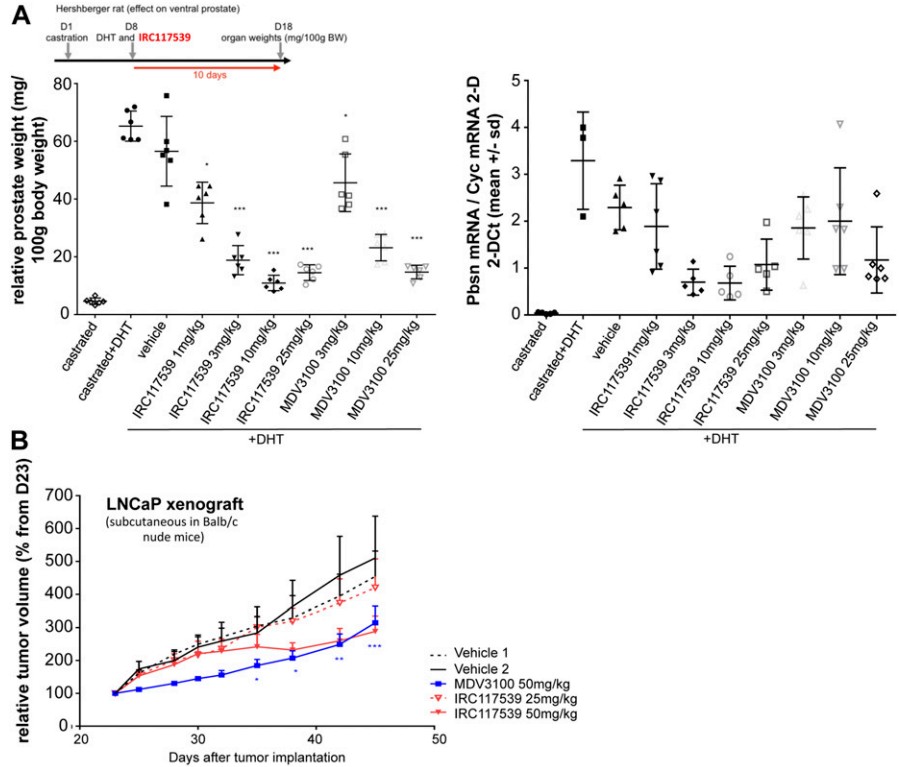

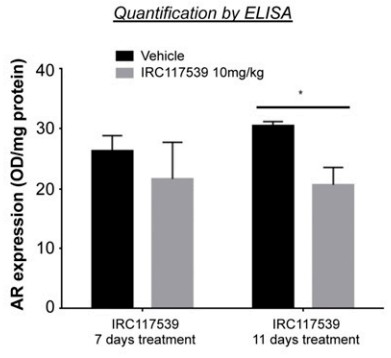

**Figure 5.** **In vivo efficacy of the IRC117539 compound.**
**(A)** Effect of IRC117539 on ventral prostate weight in a Hershberger rat after 10 d of treatment in vivo at increasing doses (left panel). Transcript levels of probasin (Pbsn), a protein abundantly expressed in prostate epithelia, were determined in a Hershberger rat treated as indicated (right panel). **(B)** Decrease in tumor volume in LNCaP prostate cancer xenograft models after 46 d of treatment with IRC117539. The effect of enzalutamide (MDV3100) is also shown for comparison. **(C)**. AR expression levels in orthotopically implanted LNCaP tumors. Animals were castrated 14 d after tumor implantation, and another 14 d after castration treatments were administered for 14 d. Tumors were harvested and analyzed for AR protein expression by immunostaining and ELISA. Data information: data are presented as mean ± SEM (n > 3). Asterisks denote statistical significance (*$P < 0.05$, **$P < 0.01$, ***$P < 0.001$, using $t$ test assuming unequal variances).

therapeutic efficacy, especially in the context of CRPC, remains suboptimal.

## Discussion

Drugs currently used for the management of prostate cancer mostly act as AR antagonists but do not induce AR degradation (Taplin, 2007; Chen et al, 2008; Schweizer & Yu, 2015). Therapies aimed at inducing the degradation of AR may, thus, offer an effective way to curb prostate cancer development and progression (Ablain et al, 2011; Chen et al, 2008; Di Zazzo et al, 2016). IRC117539 is a small molecule that interacts with the LBD of AR and triggers its degradation via the proteasome. Like many other transcription factors, AR undergoes posttranslational modifications that affect its stability, subcellular localization, and transcriptional activity (Anbalagan et al, 2012; Coffey & Robson, 2012). Whereas mono-ubiquitination of AR by the ubiquitin E3 ligase RNF6 enhances its transcriptional activity in response to androgen treatment, AR poly-ubiquitination by other E3 ligases (i.e., CHIP and Mdm2) increases its turnover rate (Lin et al, 2002; Xu et al, 2009; Sarkar et al, 2014; Liu et al, 2016). AR also interacts physically with TRIM24, a RING-domain E3 ligase negatively regulating p53, which results in increased AR-dependent transcription (Patel & Barton, 2016), and TRIM24 was proposed to interact with PML (Zhong et al, 1999). AR was among the first transcription factors shown to be SUMOylated (Poukka et al, 2000), and a very recent work from Zhang et al shows that lack of AR sumoylation leads to epidydimal dysfunction and infertility in mice (Zhang et al, 2019). We found that treatment with IRC117539 induces AR interaction with PML and promotes its massive conjugation by SUMO2/3. In particular, drug-induced change in AR/SUMO2/3 PLA signals was quite remarkable (Fig 3A). PML is known to interact with multiple proteins in response to stress, favoring their SUMO conjugation (Guo et al, 2014; Sahin et al, 2014a, 2014c, 2014d). The latter, especially by SUMO2/3, is often followed by SUMO-triggered poly-ubiquitination and degradation (Sahin et al, 2014c). Indeed, IRC117539 also enhanced the ubiquitination of AR and its association with proteasomes and ultimately triggered its degradation in a global sumoylation-dependent manner (Fig 3C). Any causal effect between IRC117539-induced AR sumoylation and its degradation remains to be demonstrated. PML association, most likely mediated via seven SIM on AR (Sahin et al, 2014c) may facilitate AR sumoylation and degradation, similar to the recently uncovered mechanism of therapy-induced catabolism of the HTLV1 Tax oncoprotein (Sahin et al, 2014c; Dassouki et al, 2015).

We noted an off-target effect of the IRC117539 compound as a weak proteasome inhibitor. This was supported by three key observations: (1) global accumulation of ubiquitin and SUMO conjugates, which may reflect inhibition of proteolysis, (2) massive stabilization of short-lived proteins (i.e., p53, Mdm2), (3) inhibition of proteasome tryptic and chymotryptic activities. Critically, IRC117539's off-target activity as a proteasome modulator was observed in both AR-positive (LNCaP and VCaP) and AR-negative (PC3 and Du145) prostate cancer cells alike. Other primary and transformed cell lines (WI38 and HeLa) also displayed similar signs of proteasome inhibition upon exposure to IRC117539. However, the compound impaired survival exclusively in AR-positive cell lines

(LNCaP, VCaP, and 22Rv1), strongly suggesting that loss of viability is uncoupled from proteasome inhibition and most likely a consequence of drug-induced elimination of a driver oncoprotein (AR). Indeed, compared with commonly used proteasome inhibitors such as MG132, IRC117539's impact on the ubiquitin–proteasome system remains rather weak, not enough to trigger significant ER stress or cause cell death, but possibly sufficient to diminish the rate of AR turnover at the proteasome in vivo (see below). In that respect, we have observed that simultaneous treatment with PD169316, a potent small molecule proteasome activator, drastically enhanced IRC117539-induced AR degradation ex vivo. Importantly, IRC117539 achieves full AR degradation at doses comparable with those required for AR inhibition and survival impairment, whereas a log-fold increase in concentration is needed for full proteasome inhibition (Fig S6D).

Remarkably, IRC117539 induces loss of viability in AR-dependent, yet androgen-insensitive 22Rv1 cell line. 22Rv1 cells express several AR isoforms: a full-length version with duplicated exon 3 that retains the LBD (for simplicity, will be referred to as full-length AR) and two truncated isoforms lacking the C-terminal LBD that are constitutively active (Dehm et al, 2008; Cunningham & You, 2015). Although the full-length isoform is still capable of interacting with androgens, co-expression of truncated isoforms results in uncoupling of androgen binding from AR activity, thus conferring androgen insensitivity to 22Rv1 cells. Truncated AR may be insensitive to androgens and can drive maximal gene expression in their absence, yet still be sensitive to IRC117539 through hetero-dimerization with a full-length AR. Heterodimerizations may precipitate proteasome-mediated destruction of the truncated variant along with the full-length form after IRC117539 binding to the full-length partner, co-sumoylation, and co-recruitment into PML NBs. Indeed, we observed that IRC117539 achieved destruction of both full-length and truncated AR in 22Rv1 cells (Fig 2A, note that we detected one major truncated variant). Finally, we cannot rule out the possibility that IRC117539-induced changes in AR sumoylation and stability may proceed independently of drug's interaction with AR LBD, for example, by involving global changes in protein sumoylation or turnover. In any case, IRC117539 is a remarkable compound that induces selective death of AR-dependent prostate cancer cells regardless of their androgen sensitivity, and may especially be clinically relevant for certain cases of CRPC.

How exactly IRC117539 impairs proteasome function remains to be determined. The compound may directly bind and destabilize defined proteasome subunits, impair proper proteasome assembly or control subunit transcription. Dexamethasone, a glucocorticoid receptor agonist, was recently shown to modify the subunit composition and enzymatic activity of proteasomes (Combaret et al, 2004), whereas cytokines such as interferons transcriptionally regulate proteasome subunit expression (Basler et al, 2013). Since AR is a transcription factor, it may alter the transcription of proteasome subunit genes; however, we were unable to unambiguously demonstrate such an effect (data not shown and Fig S6A). In contrast, IRC117539 decreased the half-lives of various proteasome subunits, notably the catalytic β7 and structural C2 of the 20S particle, which contains the tryptic-like activity. Note that the most dramatic change induced by IRC117539 is AR sumoylation (Fig 3A, AR/SUMO2/3 PLA and Fig 3B), although it may also impair

the global sumoylation cascade, which controls proteolysis and vice versa (Wang et al, 2016). We previously noted that arsenic-triggered, SUMO/ubiquitin–dependent degradation of PML or PML/RARA are much faster in vivo than in cell lines, arguing for the acquisition of changes in the efficiency of catabolic pathways with the establishment of cell lines (Lallemand-Breitenbach et al, 2001, 2008; Ferhi et al, 2016). These differences in the efficiency of sumoylation/proteasome function in vivo may explain why IRC117539 is capable of promoting AR degradation in cultured cells, whereas the molecule remains rather ineffective in vivo. We do not rule out that suboptimal pharmacokinetic properties contribute to this observation. Alternatively, IRC117539 may be metabolized in vivo into a more potent proteasome-inhibiting derivative, and we observed signs of proteasome inhibition in vivo in IRC117539-treated mice (Fig S4D). Nevertheless, the fact that IRC117539 in-duces almost total AR clearance ex vivo is remarkable and reflected in massive and selective death of prostate cancer cells that are exclusively addicted to AR. IRC117539 also shows promising effects in vivo in treatment-naïve xenograft models, comparable with anti-androgen enzalutamide. On this basis, our work will set the stage for developing IRC117539 variants towards more effective de-struction of AR by screening out its proteasome inhibitor action. In that respect, PD169316, a small molecule proteasome activator, significantly boosted IRC117539's potency in inducing AR degra-dation. Our work raises the possibility to target AR-dependent, anti–androgen-resistant tumors in CRPC, a clinical challenge for which IRC117539-derived drugs may offer an attractive treatment strategy.

# Materials and Methods

### Cell culture, treatments, Western blot, and PLAs

LNCaP, 22Rv1, PC3, and Du145 cells were cultured in 10% FCS RPMI, and MiaPaca2, VCaP, Hela, and WI38 cells were cultured in 10% FCS DMEM media (Gibco) with additional nutrients as stated in ATCC guidelines. For survival assays, cells were treated with 10 $\mu$M IRC117539 for up to 6 d. For AR degradation, AR-dependent gene expression, and PLAs, LNCaP cells were treated with indicated doses (100, 300, or 1,000 nM) of IRC117539 for indicated times (up to 48 h). MG132 (EMD Millipore) was used at a concentration of 2 $\mu$M for 24 h before cell lysis and Western blot. Rabbit polyclonal anti-human AR (N-20), p53 (DO-1), and Mdm2 (C-18) antibodies were from Santa Cruz; antibodies for proteasome subunits (PSMB7: ab154745, S7: ab3322, C2: ab3325, and PSMC3: ab171969) were from Abcam; anti-CHOP (L63F7) and anti-BiP (C50B12) antibodies were from Cell Signaling Technologies; and anti-vinculin (7F9) antibody was from Santa Cruz. All other antibodies were described previously (Sahin et al, 2014c). PLAs were performed to detect AR physical interactions with the proteasome (20S subunit), SUMO1, SUMO2/3, ubiquitin, and PML as detailed previously (Sahin et al, 2014c, 2016). Cycloheximide was from Sigma-Aldrich (used at 50 $\mu$g/ml), PD169316 (proteasome activator) was from Sigma-Aldrich (used at 10 $\mu$M), and ML792 (sumoylation inhibitor) was from MedKoo Biosciences (used at 1 $\mu$M). To analyze sumoylation and ubiquitination of endogenous or transfected AR, immunoprecipitations were performed as described

previously in detail (Sahin et al, 2014c). During immunoprecipitations, cells (including control) were co-treated with MG132 (2 $\mu$M) to stabilize modified AR forms.

### Real-time PCR analyses

Expression levels of human *FKBP5*, *PSA*, *NKX3-1*, and *AR* transcripts were determined by quantitative PCR using TaqMan probe-based gene expression analysis (Applied Biosystems), where a *Cyclophilin (Cyc)* probe was used as an internal control.

Expression levels of human *PSMA3*, *PSMB1*, *PSMB6*, *PSMB7*, *C2*, and *PSMC3* proteasome subunit transcripts were determined using a TaqMan array for human ubiquitin–proteasome–dependent proteolysis kit from Thermo Fisher Scientific (4414198).

### Cell-based proteasome activity assays

Proteasome's tryptic, chymotryptic, and caspase-like activities were analyzed in LNCaP cells after a 24-h-long treatment with increasing doses of IRC117539, using the proteasome activity assay kit from Promega (G8660) according to the manufacturer's in-structions. Briefly, LNCaP cells, seeded on 96-well plates, were treated (or untreated) with IRC117539 for 24 h before incubation with relevant fluorescent peptide substrates (Succ-LLVY-AMC, Z-LRR-AMC, or Z-nLPnLD-AMC), the cleavage of which was moni-tored using a luminometer. The latter is reflective of cellular proteasome enzymatic activities (chymotryptic, tryptic, and cas-pase, respectively).

### Fluorescence resonance energy transfer (FRET) for AR-ligand binding

FRET was determined using LanthaScreen TR-FRET AR Coactivator Assay and LanthaScreen TR-FRET AR T887A Coactivator Assay from Thermo Fisher Scientific according to the manufacturer's in-structions. Assays were run in antagonist mode. Cyproterone ac-etate or IRC117539 were incubated up to 5 h with 5 nM AR LBD or 5 nM AR T887A-LBD, 500 nM fluorescein D11FxxLF, 5 nM Tb anti-GST an-tibody, and 5 nM dihydrotestosterone. Curves were generated using XLfit software from IDBS.

### In vivo treatments and analyses

Mouse handling was performed in accordance with established institutional guidance and approved protocols from the Comité Régional d'Ethique Expérimentation Animale n°4, which enforces the EU 86/609 directive.

The Hershberger bioassay serves to assess the potency of androgens and anti-androgens in castrated rats upon treatment and is based on determining their effect on ventral prostate weight, an androgen-dependent tissue. Castrated rats were treated with IRC117539, along with dihydrotestosterone, for 10 d before sacrifice. *Probasin* mRNA levels were determined by quantitative PCR using TaqMan probe-based gene expression analysis (Applied Biosystems), where *Cyclophilin (Cyc)* served as internal control.

LNCaP xenografts were transplanted subcutaneously into Balb/c nude mice and tumor volume was recorded for up to 50 d upon treatment with IRC117539 or enzalutamide (MDV3100). For orthotopic model, LNCaP tumor cells were injected into ventral prostate of Balb/c nude mice. PSA levels were determined in peripheral blood using ELISA (PSA kit ref: EIA4105, DRG International). Once elevation of PSA levels was observed, the mice were castrated. PSA levels decreased and started increasing again approximately 14 d after castration, mimicking CRPC state. Animals were then treated with IRC117539 for 14 d, tumors were harvested, and AR levels were measured in tumors either by IHC (formalin-fixed, paraffin-embedded samples were cut into 3-$\mu$m slices, heat demasking was applied, and AR was stained using anti-AR [ref SC-7305; Santa Cruz] at 1/100 dilution) or ELISA (NR Sandwich AR; Active Motif).

## Supplementary Information

## Acknowledgements

Marion Huchet, an important contributor to this study passed away during the revision process. We acknowledge her work as, sadly, she cannot be an author of this publication. The authors wish to thank Luce Bruetschy, Denis Carré, Christelle Charnet, Sophie Chaumeron, Raphael Delille, Anne Espirito, Shari Dini Mohamed, and Frédéric Scaerou at IPSEN. We thank the imaging core facility of the Institut Universitaire d'Hématologie for microscopy analyses. The core facility is supported by grants from the Association Saint-Louis, Conseil Regional d'Ile-de-France, and the Ministère de la Recherche. We thank Dr Nathan Lack (Vancouver Prostate Center, Vancouver and Koç University, Istanbul) for prostate cancer cell lines. This work was supported by a fund from IPSEN, which covered US salary and partly by a European Molecular Biology Organization (EMBO) Small Grant (to U Sahin, 2019). EMBO Young Investigator Programme Installation Grant (IG3336) supports U Sahin laboratory.

### Author Contributions

S Auvin: formal analysis and investigation.
H Öztürk: formal analysis, investigation, and writing—review and editing.
YT Abaci: formal analysis and investigation.
G Mautino: formal analysis and investigation.
F Meyer-Losic: formal analysis, investigation, and writing—review and editing.
F Jollivet: formal analysis and investigation.
T Bashir: formal analysis, investigation, and writing—review and editing.
H de Thé: formal analysis, supervision, and writing—original draft, review, and editing.
U Sahin: conceptualization, formal analysis, supervision, validation, investigation, and writing—original draft, review, and editing.

### Conflict of Interest Statement

S Auvin, G Mautino, F Meyer-Losic, and T Bashir are employees of IPSEN that also paid US salary. This study was funded by IPSEN.

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
