## [Reviewer comments · Life Science Alliance]

Life Science Alliance

A molecule inducing androgen receptor degradation and selectively targeting prostate cancer cells

Serge Auvin, Harun Öztürk, Yusuf Abaci, Gisele Mautino, Florence Meyer-Losic, Florence JOLLIVET, Tarig Bashir, Hugues de Thé, and Umut Sahin

DOI: <https://doi.org/10.26508/lsa.201800213>

Corresponding author(s): Umut Sahin, Bogazici University

Review Timeline:

Submission Date:	2018-10-11
Editorial Decision:	2018-11-19
Revision Received:	2019-06-07
Editorial Decision:	2019-07-15
Revision Received:	2019-07-18
Accepted:	2019-07-19

Scientific Editor: Andrea Leibfried

Transaction Report:

November 19, 2018

Re: Life Science Alliance manuscript #LSA-2018-00213

Prof. Umut Sahin
Bogazici University
Department of Molecular Biology and Genetics
Center for Life Sciences and Technologies
Istanbul 34342
Turkey

Dear Dr. Sahin,

Thank you for submitting your manuscript entitled "Interference with proteasomes blunts growth inhibition resulting from androgen receptor degradation" to Life Science Alliance. The manuscript was assessed by expert reviewers, whose comments are appended to this letter.

As you will see, the reviewers find the identification of a new AR inhibitor important. However, they also point out some weaknesses in your dataset. Reviewer #1 thinks that more rigor is needed for analyses and data presentation, and that dose-titration curves should be provided as well as addressing that some results may be due to off-target effects. This reviewer also points out that some of your conclusions need down-toning. Reviewer #2 thinks that the various cell lines analyzed should be used consistently in all assays, and that IC50 values need to get reported. This reviewer also notes that the two parts of your manuscript are too disconnected, and we agree with this view.

We would like to ask you to submit a revised version of your manuscript, addressing the above-mentioned issues. Maybe you have data at hand that may help connecting the two parts of your manuscript better, or, alternatively, you can perhaps find a good way to do so by changing the text.

Thank you for this interesting contribution to Life Science Alliance. We are looking forward to receiving your revised manuscript.

Sincerely,

- A letter addressing the reviewers' comments point by point.
- An editable version of the final text (.DOC or .DOCX) is needed for copyediting (no PDFs).
- High-resolution figure, supplementary figure and video files uploaded as individual files: See our detailed guidelines for preparing your production-ready images, <http://life-science-alliance.org/authorguide>
- Summary blurb (enter in submission system): A short text summarizing in a single sentence the study (max. 200 characters including spaces). This text is used in conjunction with the titles of papers, hence should be informative and complementary to the title and running title. It should describe the context and significance of the findings for a general readership; it should be written in the present tense and refer to the work in the third person. Author names should not be mentioned.

B. MANUSCRIPT ORGANIZATION AND FORMATTING:

Full guidelines are available on our Instructions for Authors page, <http://life-science-alliance.org/authorguide>

Reviewer #1 (Comments to the Authors (Required)):

In this MS, the authors report the discovery of IRC117539 that purportedly both inhibit AR activity and target it for degradation. I think the results of this molecule is important and should be published. However, the current MS does not convincingly show that inhibits tumor growth through AR inhibition, and that it either degrades AR or it inhibits proteasomes at relevant in vivo exposure.

1. In general, please add drug concentrations to figure legends or to figure itself (Fig 2, Fig 3) to make it easier to assess.
2. In Fig. 1, it is unexpected and intriguing that IRC117539 is toxic to 22Rv1 cells, which is intrinsically resistant to enzalutamide due to expression of ARv7 variant. This suggests that IRC117539 has either 1. Off target activity, or 2. Inhibit AR in a LBD independent manner. Please delineate.
3. Several compounds are purported to degrade AR, including some that have gone onto trials. All of these degrade AR at a log-fold increase in concentration compared to that required to inhibit AR and they do not degrade AR in vivo and trials are not promising. Only fig 1 looks at a dose-response curve. Fig. 2 and Fig 3 also needs a dose-titration curve to compare dose of AR inhibition vs dose of proteasome inhibition and AR degradation.
4. The first timepoint is 24 hours. This is way too long for true changes in protein stability. Effect of MG132 on unstable proteins such as p53 can be appreciated by 30 minutes. Studying 24 and 48 hours would include many non-direct secondary effects.
5. Fig 4B, delayed effect is likely not due to AR inhibition. Is the high dose toxic to mice and are they losing weight?
6. Fig 4C is unconvincing for any AR degradation. For example, enzalutamide which does not actively degrade AR also significantly decrease AR nuclear staining in xenograft tumors (more than shown for IRC117539 here). Authors should just say that their drug does not significantly decrease AR protein in vivo.
7. For Fig 5, the dose tested to stabilize proteins is often much higher than therapeutic dose. So one cannot imply that this is happening in patients in vivo.

Reviewer #2 (Comments to the Authors (Required)):

The manuscript describes a novel rationally designed AR inhibitor which promotes AR degradation, IRC117539. The authors provide evidence that degradation is proteasomally dependent. The latter observation, based on MG132 treatment, is most clear at the 24-hour timepoint in Fig. 2B. The authors should comment on the less significant effect at 48 hours. Is this because the cells are dying or due to some other factor? Having established a high degree of specificity in a panel of PCa cell-lines in Fig.1 the manuscript then reports some off-target effects including inhibition of proteolysis as assayed through the accumulation of ubiquitin and Sumo conjugates. These data are shown in figure 3 and at this point some cell-lines are used which were not used in figure 1 including HeLa and WI38 cells. It would be helpful to know whether these effects are also observed in the AR-negative PCa cell-lines used in figure 1. It would be reciprocally be helpful to know what the IC50 values are for the two lines used in figure 3 (HeLa and WI38). In the context of figure 3 it would be helpful to know whether the effect of accumulating ubiquitin and Sumo conjugates is to trigger the activation of unfolded protein response pathways and whether that in turn is leading to cell death - markers such as ATF4 and CHOP could be used. It would also be helpful to know whether some or all of these changes are observed in harvested tumour tissue from subsequent xenograft experiments. Overall the manuscript is interesting but somewhat disjointed in its current

form. A rather specific drug has some off-target effects but it is not entirely clear which dominate to determine the growth effects on cancer cells and consequently how significant the off-target effects might prove to be. Ensuring that many of the reported measurements are made across the full range of models used in the paper will help.

November 19, 2018

Re: Life Science Alliance manuscript #LSA-2018-00213

Prof. Umut Sahin
Bogazici University
Department of Molecular Biology and Genetics
Center for Life Sciences and Technologies
Istanbul 34342
Turkey

Dear Dr. Sahin,

Thank you for submitting your manuscript entitled "Interference with proteasomes blunts growth inhibition resulting from androgen receptor degradation" to Life Science Alliance. The manuscript was assessed by expert reviewers, whose comments are appended to this letter.

As you will see, the reviewers find the identification of a new AR inhibitor important. However, they also point out some weaknesses in your dataset. Reviewer #1 thinks that more rigor is needed for analyses and data presentation, and that dose-titration curves should be provided as well as addressing that some results may be due to off-target effects. This reviewer also points out that some of your conclusions need down-toning. Reviewer #2 thinks that the various cell lines analyzed should be used consistently in all assays, and that IC50 values need to get reported. This reviewer also notes that the two parts of your manuscript are too disconnected, and we agree with this view.

We would like to ask you to submit a revised version of your manuscript, addressing the above-mentioned issues. Maybe you have data at hand that may help connecting the two parts of your manuscript better, or, alternatively, you can perhaps find a good way to do so by changing the text.

Thank you for this interesting contribution to Life Science Alliance. We are looking forward to receiving your revised manuscript.

Sincerely,

Andrea Leibfried, PhD
Executive Editor
Life Science Alliance
Meyerhofstr. 1
69117 Heidelberg, Germany

Reviewer #1 (Comments to the Authors (Required)):

In this MS, the authors report the discovery of IRC117539 that purportedly both inhibit AR activity and target it for degradation. I think the results of this molecule is important and should be published. However, the current MS does not convincingly show that inhibits tumor growth through AR inhibition, and that it either degrades AR or it inhibits proteasomes at relevant in vivo exposure.

We thank the reviewer for appreciating the importance of our data on IRC117539. We have now extensively modified the manuscript, rigorously studied IRC117539 in various AR(+) and AR(-) prostate cancer (PC) cell lines, and present more data that support that IRC117539 exerts its growth inhibitory effect mainly through AR degradation, rather than proteasome inhibition.

For this purpose we systematically tested IRC117539 on 8 different cell lines: 3 AR-dependent prostate cancer cells (LNCaP, VCaP and 22Rv1), 2 AR negative prostate cancer cells (PC3 and Du145), 3 other cell types whose proliferation is not AR dependent (MiaPaca2 pancreatic cancer, HeLa cervical cancer, and WI38 primary human fibroblasts). Importantly, 6 of these exhibited similar signs of proteasome inhibition when treated with IRC117539 (LNCaP, VCaP, PC3, Du145, HeLa, WI38) (Fig 4A, Fig S4B, Fig S5). We did not observe proteasome inhibition in 22Rv1 cells (Fig S4C), and could not obtain data in MiaPaca2 cells due to technical challenges. Critically, IRC117539 consistently induced loss of viability only in AR-dependent PC cells (LNCaP, VCaP and 22Rv1) but not in others (Fig 1A, Fig 1B, Fig S1B). Of note, while survival of AR(-) PC3, Du145, HeLa, WI38 cells (which display proteasome inhibition) was not affected, AR(+) 22Rv1 cells lose viability (despite lack of proteasome inhibition). These observations strongly argue that loss of viability is uncoupled from proteasome inhibition, and that IRC117539's off-target effect on proteasomes per se does not lead to cell death.

In line with the notion that AR degradation initiates PC cell death, we could also successfully show that IRC117539 induces AR degradation also in VCaP and 22Rv1 cells (in the original manuscript, we had shown this only in LNCaP cells).

Thus, a) correlation between 'AR loss' and 'loss of viability' in all 3 AR(+) PC cells, b) uncoupling of proteasome inhibition from mortality, strongly argue that IRC117539's growth inhibitory effect on PC cells stems mainly from AR degradation, rather than off-target effects.

In fact, IRC117539 only mildly affects the proteasomes: IRC117539-induced accumulation of ubiquitinated proteins is mild when compared with a standard proteasome inhibitor such as MG132 (Fig S6C). IRC117539 does not cause significant ER distress (Fig S6C), and its IC_{50} on proteasomes is much higher than its IC_{50} on AR-dependent PC cell survival (for LNCaP cells, $IC_{50(survival)}$: 614 nM and $IC_{50(proteasome)}$: 878 μ M) (Fig 1A and 4C).

This unexpected weak off-target activity, though not inhibitory on cell survival, may antagonize drug's primary function and impede AR degradation in vivo.

Indeed, we observed signs of proteasome inhibition at relevant in vivo exposure in mice (Fig S4D). In addition, at least ex vivo, we now show that PD169316, a small molecule proteasome activity booster, significantly enhances IRC117539-induced AR degradation (Fig 2D). Nevertheless, we agree with the reviewer and thus toned-down some of our conclusions drawn from in vivo work and merely speculated as to why IRC117539 may not be as effective in vivo in inducing AR degradation (i.e. differences in the efficiency of catabolic pathways in cell lines versus in vivo; suboptimal pharmacokinetics; metabolic derivatives with more potent proteasome-inhibiting activity, etc as discussed on page 15 in Discussion).

We also acknowledge that the initial manuscript may have appeared as if it contained two parts: on one hand, description of IRC117539 as a novel AR-degrading drug; and on the other hand, modulation of global proteolysis as an off-target effect, which may also concern numerous FDA-approved drugs. In order to ensure fluidity and better connect these two parts, we have now 1) shown that AR loss, but not proteasome inhibition, strongly correlates with IRC117539's growth-inhibitory effect, 2) focused exclusively on 'reporting IRC117539 as a novel AR-degrading molecule' and explored its mechanism of action in further detail (Fig 2, Fig 3 and FigS3), 3) based on our discussions with the editor, removed data concerning other FDA-approved drugs (previous Fig 5).

Concerning the mechanism of IRC117539-induced AR degradation, we now corroborated our initial proximity ligation analyses (PLA) with robust immunoprecipitation data. IRC117539 clearly induces massive SUMO(2/3)ylation of endogenous AR in LNCaP cells, which seems to be a prerequisite for its degradation (Fig 3B and 3C). This catabolic pathway is intriguingly reminiscent of therapy-induced PML/RARA degradation in acute promyelocytic leukemia and therapy-induced Tax oncoprotein degradation in adult T-cell lymphoma. In both cases, drug-induced clearance of these oncoproteins results in growth inhibition of cancer cells.

1. In general, please add drug concentrations to figure legends or to figure itself (Fig 2, Fig 3) to make it easier to assess.

Drug concentrations are now indicated on both main and supplementary figures and/or legends. Please note that Fig 2 now shows data from LNCaP, VCaP and 22Rv1 cells and Fig 4 (old Fig 3) now shows data on LNCaP, VCaP, PC3 and Du145 cells.

2. In Fig. 1, it is unexpected and intriguing that IRC117539 is toxic to 22Rv1 cells, which is intrinsically resistant to enzalutamide due to expression of ARv7 variant. This suggests that IRC117539 has either 1. Off target activity, or 2. Inhibit AR in a LBD independent manner. Please delineate.

We thank the reviewer for raising this interesting point. 22Rv1 cells are indeed AR-dependent, yet androgen insensitive because they express the truncated AR variant (ARv7). AR dimerizes to initiate target gene transcription. Several modes of AR dimerization have been proposed, including dimerization through interactions between N-terminus/C-terminus and dimerization through DNA-binding domain

(Centenera et al., 2008). Based on any of these modes, it is conceivable that ARv7 may dimerize with a full-length (FL) AR, which is also expressed in 22Rv1 cells (Fig 2A and Cunningham and You, 2015; Dehm et al., 2008). Indeed, truncated AR variants were recently shown to not only homodimerize with each other but also heterodimerize with full-length AR (Xu et al., 2015). In principle, IRC117539 can bind to LBD of FL AR forming such a heterodimer, and this can subsequently initiate destruction of the truncated partner via SUMOylation and ubiquitination in trans by SUMO/ubiquitin ligases, or via co-recruitment into PML NBs. Indeed, we could strikingly show that IRC117539 initiates degradation of both FL AR and ARv7 in 22Rv1 cells (Fig 2A). ARv7, though insensitive to androgens or enzalutamide, may thus still be sensitive to IRC117539-induced degradation through heterodimerization with FL AR. This could explain androgen/enzalutamide insensitivity, yet IRC117539 sensitivity of 22Rv1 cells.

For the reasons stated earlier, we think that off-target drug activity plays minimal role in impairment of 22Rv1 survival. In addition, Fig 1C also clearly shows that IRC117539 interacts with AR LBD. We acknowledge the need for future experiments to explore how ARv7 is degraded, thank the reviewer for raising this issue and believe that our data now raise the possibility to target a subset of AR(+), anti-androgen resistant CRPC tumors, which remain a clinical challenge. A paragraph is added in the discussion to underline this intriguing point.

3. Several compounds are purported to degrade AR, including some that have gone onto trials. All of these degrade AR at a log-fold increase in concentration compared to that required to inhibit AR and they do not degrade AR in vivo and trials are not promising. Only fig 1 looks at a dose-response curve. Fig. 2 and Fig 3 also needs a dose-titration curve to compare dose of AR inhibition vs dose of proteasome inhibition and AR degradation.

*This is a valid point and we appreciate the value of proposed comparisons. We have now added dose-response curves for AR degradation (Fig 2B) and proteasome inhibition (Fig 4B and 4C) for comparison with AR inhibition and cell survival curves in Fig 1. Importantly and contrary to previously reported compounds, IRC117539 achieves full AR degradation at doses comparable to those required for AR inhibition and survival impairment (around or less than 1000 nM). On the other hand, a log-fold increase in concentration is needed for proteasome inhibition, at least *ex vivo* in cultured cells (Fig S6D).*

4. The first timepoint is 24 hours. This is way too long for true changes in protein stability. Effect of MG132 on unstable proteins such as p53 can be appreciated by 30 minutes. Studying 24 and 48 hours would include many non-direct secondary effects.

We agree with the reviewer. Therefore, we now show data and statistical analyses for 6, 12, 24 and 48 hrs. We observe significant AR degradation as early as 6 hrs upon treatment, which reaches near completion by 12 hrs in LNCaP cells (Fig 2A and 2B). In Fig 4, we also studied accumulation of ubiquitinated proteins at earlier

time points. Consistent with IRC117539 being a mild inhibitor of the proteasome, a) significant accumulation was observed only at later time points (upon prolonged treatment) in most cell lines (Fig 4), b) the effect of MG132 on unstable ubiquitin conjugates was more pronounced than that of IRC117539 (Fig S6C).

5. Fig 4B, delayed effect is likely not due to AR inhibition. Is the high dose toxic to mice and are they losing weight?

In our hands, the compound does not seem to be toxic per se (please see below the body weight curves, days plotted against body weight), the vehicle is a bit viscous so we tend to lose animals during the treatment period due to oral gavage. In this specific experiment mentioned by the reviewer (now, Fig 5B), we lost 1 animal in the vehicle group and 3 in the 50 mg/kg dose group. The vehicle is different for MDV3100. Otherwise, the animals treated with IRC117539 look similar to the vehicle-treated ones; we observe no clinical signs other than those related to the gavage.

We agree with the reviewer that the kinetics of IRC117539's effect is different from that of MDV3100. We tend to see this as a difference in the mechanism of action. Inhibition of AR pathway is less strong with IRC117539, which requires AR degradation, which, in turn, takes time. In addition, we also take into account the different pharmacokinetics of these molecules, as it simply takes more time for IRC to reach and keep inhibitory concentrations in tumors.

6. Fig 4C is unconvincing for any AR degradation. For example, enzalutamide which does not actively degrade AR also significantly decrease AR nuclear staining in xenograft tumors (more than shown for IRC117539 here). Authors should just say that their drug does not significantly decrease AR protein in vivo.

We have now deleted the sentence in the initial manuscript “In castration resistant orthotopic LNCaP xenograft models, IRC117539 treatment was able to induce a transient and moderate AR decrease (about 30% decrease as determined by immunohistochemistry and ELISA, Fig 4C)”; and now added in the new version (page 10/11) *“In castration resistant orthotopic LNCaP xenograft models, IRC117539 treatment did not significantly decrease AR protein levels (as determined by immunohistochemistry and ELISA, Fig 5C) and neither...”*

7. For Fig 5, the dose tested to stabilize proteins is often much higher than therapeutic dose. So one cannot imply that this is happening in patients in vivo.

We agree with the reviewer. Following our discussions with the editor and in order to improve both the flow and the integrity of the manuscript, we have now removed old Fig 5, and decided to focus entirely on IRC117539 and its mechanism of action.

Reviewer #2 (Comments to the Authors (Required)):

We thank this reviewer for expressing his/her interest in our manuscript, which we have considerably improved. We have now used the panel of prostate cancer (PC) cell lines introduced in Fig 1A in a more consistent manner; obtained robust data on AR degradation in 3 different PC lines; further explored the mechanism underlying AR degradation; studied IRC117539's off-target effect in most of the cell lines shown in Fig 1A; added dose-response curves/or IC₅₀ values for AR degradation (Fig 2B) and proteasome inhibition (Fig 4B and 4C) for comparison with AR inhibition and cell survival curves in Fig 1.

We agree with the reviewer that the initial manuscript may have appeared disjointed. Our data now strongly argue that AR degradation dominates to determine IRC117539's growth inhibitory effect on cancer cells, and proteasome inhibition per se does not suffice to cause loss of viability. To ensure manuscript's integrity: 1) we show that IRC117539-induced loss of viability is uncoupled from proteasome inhibition, but consistently correlates with AR degradation, 2) we focus exclusively on reporting IRC117539 as a novel molecule, explore its mechanism of action; and based on our discussions with the editor, we removed data concerning other FDA-approved drugs (previous Fig 5).

The manuscript describes a novel rationally designed AR inhibitor which promotes AR degradation, IRC117539. The authors provide evidence that degradation is proteasomally dependent. The latter observation, based on MG132 treatment, is most clear at the 24-hour timepoint in Fig. 2B. The authors should comment on the less significant effect at 48 hours. Is this because the cells are dying or due to some other factor?

The reviewer is correct in that in previous Fig 2B, the effect of MG132 at 48 hrs looked unimpressive. This is due to the technical difficulty of treating cells with MG132 for 48 hrs due to toxicity: in order to minimize toxicity, MG132 treatment was performed for 24 hrs only. 48 hour long treatments were not feasible. Thus, for the 48 hr data point, MG132 was added only 24 hrs prior to lysis. At this point, cells

were already exposed to IRC117539 for a duration of 24 hrs and most of AR was already degraded (our new data indicate that IRC117539-induced AR degradation is quite fast and reaches near completion by 24 hrs, as shown in Fig 2A, 2B and 2C). In addition, IRC117539 treatment induces death in LNCaP cells (Fig 1B), adding MG132 at this point only introduced further toxicity. We performed this experiment several times, also including an earlier time point at 6 hrs, and consistently observed that MG132 prevented IRC117539-induced AR degradation as long as it was present during IRC117539 treatment (please see new Fig 2C). Below, we show the uncropped gel to the reviewer, but for simplicity we chose to omit the 48 hrs time point in the paper. Finally, in line with proteasome-dependent degradation of AR, we could show that PD169316, a small molecule proteasome activity booster, significantly enhanced IRC117539-induced AR degradation (Fig 2D).

Having established a high degree of specificity in a panel of PCa cell-lines in Fig.1 the manuscript then reports some off-target effects including inhibition of proteolysis as assayed through the accumulation of ubiquitin and Sumo conjugates. These data are shown in figure 3 and at this point some cell-lines are used which were not used in figure 1 including HeLa and WI38 cells. it would be helpful to know whether these effects are also observed in the AR-negative PCa cell-lines used in figure 1.

We agree with the reviewer and appreciate the value of the proposed experiment. New Figs 4A and S4B now show drug's effect on global proteolysis in two of the AR-negative PC lines (PC3 and Du145) mentioned in Fig 1A, as well as in two AR-positive PC lines (LNCaP and VCaP). We could not obtain further data in MiaPaca2 cells due to technical reasons (contamination).

To answer the reviewer's question, both of the AR-negative PC lines (PC3 and Du145) displayed accumulation of global ubiquitin/SUMO conjugates.

We did not observe proteasome inhibition in 22Rv1 cells (Fig S4C). Critically, IRC117539 consistently induced loss of viability only in AR-dependent PC cells (LNCaP, VCaP and 22Rv1) but not in others (Fig 1A, Fig 1B, Fig S1B). Of note, while survival of AR(-) PC3, Du145, HeLa, WI38 cells (all of which display proteasome inhibition) was not affected, AR(+) 22Rv1 cells lose viability (despite lack of apparent proteasome inhibition). These observations strongly argue that loss of viability is uncoupled from proteasome inhibition, and that IRC117539's off-target effect on proteasomes per se does not lead to cell death.

In line with the notion that AR degradation initiates PC cell death, we could also successfully show that IRC117539 induces AR degradation also in VCaP and 22Rv1 cells (in the original manuscript, we had shown this only in LNCaP cells).

Thus, a) correlation between 'AR loss' and 'loss of viability' in all 3 AR(+) PC cells, b) uncoupling of proteasome inhibition from mortality, strongly argue that IRC117539's growth inhibitory effect on PC cells stems mainly from AR degradation, rather than off-target effects.

It would be reciprocally be helpful to know what the IC50 values are for the two lines used in figure 3 (HeLa and WI38).

We agree with the reviewer and now show the IC50 values for HeLa and WI38 cells in Fig S1B. As in other AR(-) cell lines used in Fig 1A, survival of these two cell types was also only slightly affected by the compound.

Kindly note that to ensure consistency, in new Fig 4 (old Fig 3), we show data from two of the AR-negative PC lines (PC3 and Du145) and two of the AR-positive PC lines (LNCaP and VCaP) introduced earlier in Fig 1A, We thus moved HeLa and WI38 data to Fig S5.

In the context of figure 3 it would be helpful to know whether the effect of accumulating ubiquitin and Sumo conjugates is to trigger the activation of unfolded protein response pathways and whether that in turn is leading to cell death - markers such as ATF4 and CHOP could be used. It would also be helpful to know whether some or all of these changes are observed in harvested tumour tissue from subsequent xenograft experiments.

We thank the reviewer for this suggestion and now tested two UPR markers that were available to us: CHOP and BiP (Fig S6C). Our results did not indicate a substantial rise in ER stress due to IRC117539 treatment. We observed a slight upregulation of CHOP at 24 hrs (and no upregulation of BiP), but this was drastically weaker compared to the effect of MG132, a potent ER stress inducer.

In fact, IRC117539 only mildly affects the proteasomes: IRC117539-induced accumulation of ubiquitinated proteins is mild when compared with a standard proteasome inhibitor such as MG132 (Fig S6C). Also, its IC₅₀ on proteasomes is much higher than its IC₅₀ on AR-dependent PC cell survival (for LNCaP cells, IC_{50(survival)}: 614 nM and IC_{50(proteasome)}: 878 μM) (Fig 1A and 4C).

The experiments in tumor tissues suggested by the reviewer are very valuable and would certainly improve the quality of the manuscript. Unfortunately, due to recent relocation of the principle investigator, we had no more access to previously harvested tumor tissues and follow-up experiments in xenograft models would by far exceed acceptable time frames. Thus, in order to present evidence for proteasome inhibition at relevant in vivo exposure, we treated regular Balb/c mice with IRC117539. We noted accumulation of global ubiquitin conjugates in liver tissues upon treatment with the drug (Fig S4D).

Overall the manuscript is interesting but somewhat disjointed in its current form. A rather specific drug has some off-target effects but it is not entirely clear which dominate to determine the growth effects on cancer cells and consequently how significant the off-target effects might prove to be. Ensuring that many of the reported measurements are made across the full range of models used in the paper will help.

Again, we appreciate this reviewer's interest in our manuscript. As we mentioned above in the introductory paragraph, we believe that we have now addressed most of his/her concerns:

1) by using the panel of AR(+) and AR(-) cell lines introduced in Fig 1A consistently throughout the manuscript for several key measurements (i.e. drug's effect on AR degradation, on global accumulation of ubiquitinated proteins, on cell survival).

2) by performing rigorous data analyses. We have now added dose-response curves for AR degradation (Fig 2B) and proteasome inhibition (Fig 4B, and IC_{50} in 4C) for comparison with AR inhibition and survival curves in Fig 1.

*Critically, IRC117539 achieves full AR degradation at doses comparable to those required for AR inhibition and survival impairment (around or less than 1000 nM). On the other hand, a log-fold increase in concentration is needed for proteasome inhibition, at least *ex vivo* in cultured cells (Fig S6D).*

3) by showing that IRC117539-induced PC cell death is uncoupled from its effect on global proteolysis, but strictly correlates with AR loss in AR(+) cells only.

4) by mainly focusing on 'reporting IRC117539 as a novel AR-degrading molecule'. To this end, following our discussions with the editor, we removed previous Fig 5 (data on other FDA-approved drugs). We showed that IRC117539 achieved AR degradation in numerous AR-dependent PC lines. In addition, we further explored the mechanism of IRC117539-induced AR degradation: we now corroborated our initial proximity ligation analyses (PLA) with robust immunoprecipitation data. IRC117539 clearly induces massive SUMO(2/3)ylation of endogenous AR in LNCaP cells, which seems to be a prerequisite for its degradation (Fig 3B and 3C). This catabolic pathway is intriguingly reminiscent of therapy-induced PML/RARA degradation in acute promyelocytic leukemia and therapy-induced Tax oncoprotein degradation in adult T-cell lymphoma. In both cases, drug-induced clearance of these oncoproteins results in growth inhibition of cancer cells.

*Though IRC117539's off-target effect on proteasomes does not impair cell survival, we think it may still be important to report because: a) interference with proteasomes may antagonize drug's primary function in vivo, reducing its effectiveness. We acknowledge the need for further experiments to test this in future studies, but we have included *ex vivo* data to show that boosting proteasome activity increases IRC117539's potency (Fig 2D), b) IRC117539 is a promising compound and may offer the possibility to target AR-dependent, anti-androgen-resistant tumors in CRPC (22Rv1 data in Fig 1A and 2A), and it may further be improved by screening out its undesired effect on proteasomes.*

July 15, 2019

RE: Life Science Alliance Manuscript #LSA-2018-00213R

Prof. Umut Sahin
Bogazici University
Department of Molecular Biology and Genetics
Center for Life Sciences and Technologies
Istanbul 34342
Turkey

Dear Dr. Sahin,

Thank you for submitting your revised manuscript entitled "Drug-induced androgen receptor sumoylation and degradation blunted by proteasome inhibition". As you will see, one of the original reviewers re-assessed your work and appreciates the introduced changes. A few issues remain, however. We would be happy to publish your paper in Life Science Alliance pending final revisions necessary to address the remaining concerns of the reviewer:

- please address the reviewer's comments by text changes. We agree with the reviewer that the title should also highlight that IRC117539 targets AR for degradation
- please also fill in the electronic license to publish form within our submission system

A. FINAL FILES:

-- Summary blurb (enter in submission system): A short text summarizing in a single sentence the study (max. 200 characters including spaces). This text is used in conjunction with the titles of papers, hence should be informative and complementary to the title. It should describe the context

and significance of the findings for a general readership; it should be written in the present tense and refer to the work in the third person. Author names should not be mentioned.

B. MANUSCRIPT ORGANIZATION AND FORMATTING:

Sincerely,

Andrea Leibfried, PhD
Executive Editor
Life Science Alliance
Meyershofstr. 1
69117 Heidelberg, Germany
t +49 6221 8891 502
e a.leibfried@life-science-alliance.org
www.life-science-alliance.org

Reviewer #1 (Comments to the Authors (Required)):

Overall, this is a very significant improvement from the initial manuscript. Most of my concerns are adequately addressed. I have several major suggestions:

1. I do not like the change in title. Please discuss with editor. It should highlight the discovery of new drug that induces AR degradation, in my opinion.

2. While IRC117539 and interact with AR LBD in vitro, there is no data that the induction of AR sumoylation or degradation is due to direct interaction. The binding is rather weak and many other mechanisms are possible. Again, knockdown of full-length AR in 22Rv1 cells do not significantly decrease the ARv7 protein level and the justification that they form dimers is not correct. So the drug likely also decrease ARv7 protein independently of AR full length. It is also important to note that AR sumylation sites are not in the LBD.

3. I am not satisfied with the cause/effect of between sumoylation and degradation. SAE inhibition may cause changes in degradation apparatus and is not sufficiently specific. Mutagenesis of sumoylation sites (which have been defined) would be more definitive but not necessary for this manuscript. Would change wording (including abstract) to take out cause/effect unless more experiments are done.

July 19, 2019

RE: Life Science Alliance Manuscript #LSA-2018-00213RR

Prof. Umut Sahin
Bogazici University
Department of Molecular Biology and Genetics
Center for Life Sciences and Technologies
Istanbul 34342
Turkey

Dear Dr. Sahin,

Thank you for submitting your Research Article entitled "A molecule inducing androgen receptor degradation and selectively targeting prostate cancer cells". We appreciate the introduced changes and it is a pleasure to let you know that your manuscript is now accepted for publication in Life Science Alliance. Congratulations on this interesting work.

DISTRIBUTION OF MATERIALS:

Again, congratulations on a very nice paper. I hope you found the review process to be constructive and are pleased with how the manuscript was handled editorially. We look forward to future exciting submissions from your lab.

Sincerely,

Andrea Leibfried, PhD
Executive Editor
Life Science Alliance
Meyerohofstr. 1
69117 Heidelberg, Germany
t +49 6221 8891 502
e a.leibfried@life-science-alliance.org
www.life-science-alliance.org